# Study on the Influence of Surface Treatment Process on the Corrosion Resistance of Aluminum Alloy Profile Coating

**DOI:** 10.3390/ma16176027

**Published:** 2023-09-01

**Authors:** Lei Fan, Fatao Wang, Zhouhui Wang, Xuelong Hao, Neng Yang, Denglin Ran

**Affiliations:** 1School of Mechanical Electronic & Information Engineering, China University of Mining & Technology (Beijing), Beijing 100083, China; wft1025@163.com (F.W.); wdsszxm@163.com (N.Y.); randenglin2332@163.com (D.R.); 2Key Laboratory of Coal Mine Intelligence and Robot Innovation and Application of Emergency Ministry, Beijing 100083, China; 3Fine Chemicals Group Co., Ltd., Taizhou 318020, China; office@finechemgroup.com; 4National Nonferrous Metals and Electronic Materials Analysis and Testing Center, Guobiao (Beijing) Inspection and Certification Co., Ltd., Beijing 101407, China

**Keywords:** aluminum alloy, anodizing, powder coating, electrochemistry, corrosion

## Abstract

This work focuses on different surface treatment processes of the 6061 aluminum alloy profile coatings in the construction industry, mainly including the sand powder film coating, the flat powder coating, the hard anodized film, and the ordinary heat-sealing oxidized coating. The corrosion resistance of the coated aluminum alloy in a 3.5 wt.% NaCl solution (pH 6.5–7.5) and the influence of different surface treatment processes on the corrosion resistance of different samples were studied by scanning electron microscope (SEM) and electrochemical workstation. The result shows that with the increase in corrosion time, the corrosion inhibition performance of the four coated aluminum alloy materials decreased significantly, and the order of decline is: sand powder film coating > hard anodized film > flat powder coating > ordinary heat-sealing oxidized coating. When corroded in a 3.5 wt.% NaCl solution for 2 h, the corrosion inhibition performances of the flat powder coating and ordinary heat-sealing oxidized coating are poor, while the inhibition performances of the sand powder film coating and hard anodized film are good, and the inhibition performance follows the following sequence: the sand powder film coating > hard anodized film> the flat powder coating > ordinary heat-sealing oxidized coating. When corroded in a 3.5 wt.% NaCl solution for 200 h, the corrosion inhibition performances of the sand powder film coating and the flat powder coating are poor, while the inhibition performances of hard anodized film and ordinary heat-sealing oxidized coating are good, and the inhibition performance follows the following sequence: hard anodized film > ordinary heat-sealing oxidized coating > the sand powder film coating > the flat powder coating.

## 1. Introduction

Aluminum alloys have the advantages of low density, high specific strength, excellent corrosion resistance, recyclability, and low cost. They are widely used in construction and other fields, as well as being ideal raw materials for profiles in the construction industry [1,2,3,4,5]. With the global warming issue, the occurrence of extreme weather and climate events is increasing and intensifying, especially for extreme events such as heavy rain and flooding, high temperatures, and heat waves. In this context, the frequent occurrence of extreme weather has become the new normal in China [6,7,8,9,10,11]. Therefore, higher requirements are put forward for the corrosion resistance of architectural aluminum alloy profiles. Faced with a market for aluminum alloy profiles that is becoming increasingly competitive, the way for businesses to advance is to reduce costs, enhance the manufacturing process, and continuously enhance the corrosion resistance, weather resistance, and wear resistance of the aluminum alloy profiles.

At present, the most widely used treatment methods on aluminum alloy surfaces are anodizing and powder coating [1,12,13,14]. Minhas Badar et al. [15] developed an active protective coating surface using the rapid leaching and strong barrier properties of the corrosion inhibitor combined with the anodized aluminum alloy 2024-T3, and electrochemical studies showed that the rapid and sufficient leaching of lithium led to a significant improvement in the corrosion resistance of scratch coatings. X. H. Lin et al. [16] used PEMHP (piezoelectric machine hammer peening) to study the surface modification of 6016 aluminum alloy anodizing film, including surface roughness, surface morphology, and hardness. It is found that peening parameters strongly influence surface integrity, especially line pitch. Badhotiya Gaurav Kumar et al. [17] prepared an alumina coating using oxime-modified aluminum (III) isopropanol as a sol-gel precursor. The mechanical properties of the bare, dip-coated, anodized, and coated anodized aluminum alloy samples were investigated by the slow strain rate test. It indicates that simply dip-coated surfaces have a higher ultimate tensile strength as compared with other samples. Kumar J. Jagadesh et al. [18] studied the tribological, corrosion, and mechanical properties of AISI 316L stainless steel without and with an Al_2_TiO_5_ surface coating. The results show that a twenty-fold reduction in wear rate is recorded with the coating of Al_2_TiO_5_ on the base material when compared with the uncoated counterpart. The wear rate has also decreased by 16% with the increase in coating thickness from 300 to 375 μm. The fatigue life of the coated specimens was reduced by around 12%, while their corrosion resistance increased by 20% when compared with the uncoated specimens. The Si_3_N_4_ ceramics were successfully fabricated by DLP technology using Si_3_N_4_ powder coated with Al_2_O_3_-Y_2_O_3_ sintering additives by Li Meng et al. [19].

However, there are many coating studies but less about aluminum alloys, and performance studies mainly focus on surface modification, tribology, and mechanical properties, with less research on corrosion. In addition, comparative studies between different surface treatment processes are rarely seen. In this work, the corrosion resistance of powder coatings and anodized films on the surfaces of aluminum alloy building profiles was studied [20,21,22,23,24,25,26,27,28]. Through the coupling analysis of microscopic observation and electrochemical fitting, scanning electron microscopy (SEM) and electrochemical analysis were used to investigate the corrosion behavior and corrosion resistance of the sand texture film layer, the flat powder film layer, the hard anodized, and the ordinary anodized heat-sealed hole aluminum alloy profiles in weakly corrosive electrolytes. This paper aims to compare the corrosion resistance of aluminum alloys in different surface treatment processes and provide a reference for enterprises to improve the process in the future.

## 2. Materials and Methods

### 2.1. Experimental Materials

The powder-coated aluminum alloy samples are as follows: sand powder coating (sample number 1#), flat powder coating (sample number 2#), hard anodizing film (sample number 3#), and ordinary anodized heat seal hole specimen (sample number 4#) are obtained from Guangdong Haomei New Material Co., Ltd. (Qingyuan, China). Sample dimensions 10 mm × 10 mm × 5 mm were cut from aluminum alloy for static immersion corrosion measurements. Samples of 10 mm × 10 mm × 5 mm were cut for electrochemical measurements. The samples were embedded in epoxyresin made of polytetrafluoroethylene, and a surface area of 1 cm^2^ was exposed. Before each experiment, the working electrode was mechanically abraded with emery paper beginning at 400 # to 3000 #, rinsed with double-distilled water, and dried in air.

### 2.2. Static Immersion Corrosion Experiment

The static immersion corrosion test was carried out in a 3.5 wt.% NaCl solution at 40 ± 1 °C for 2 h and 200 h of immersion (HH-ZKB constant temperature water bath). The surface morphology of the immersed samples was observed by scanning electron microscope (SEM, Hitachi, Tokyo, Japan, S-3400N) at an accelerating voltage of 15 KV and a beam current of 10 nA.

### 2.3. Electrochemical Testing

A conventional three-electrode electrochemical workstation (PARSTAT, Versa STAT 3F, AMETEK, Berwyn, PA, USA) was used for the electrochemical tests, applying different processed samples as the working electrode (WE), a platinum electrode as the counter electrode (CE), and a saturated Ag/AgCl electrode as the reference electrode (RE). The test solution was 3.5 wt.% NaCl solution. The open circuit potential (OCP) was recorded for 30 min until a steady state was observed, and the potentiodynamic polarization (PDP) test was performed in a potential range of ±500 mV (vs. OCP) at a sweep rate of 1 mV/s. The electrochemical impedance spectroscopy (EIS) test was performed under OCP with an AC frequency range of 100 kHz–10 MHz and a disturbance signal amplitude of 10 mV, and the impedance data were further analyzed and fitted using ZView2 software. The measurements were repeated three times to ensure reproducibility.

## 3. Results and Discussion

### 3.1. Micromorphology of the Coating after Corrosion

Figure 1 shows the pre-corrosion micromorphology of aluminum alloy with different surface treatment processes. The aluminum alloy (sample 1#) is coated with a layer of sand film, and the surface roughness is the largest among the four materials. Flat powder coating (sample 2#) has a smoother surface and smaller powder particle size than the sand film layer coating. A hard anodized aluminum alloy sample (sample 3#) has defects such as cracks due to the treatment process on the surface. The anodized film on the surface of ordinary anodized heat-sealed aluminum alloy (sample 4#) is unevenly sealed, and there are a large number of holes.

Figure 2a shows the microscopic corrosion morphology of the sand powder coating (sample 1#) after 200 h of immersion in the 3.5 wt.% NaCl solution. The corrosion type of the observed area is pitting corrosion; the pitting is irregular and hole-like, and the corrosion products are attached to the vicinity of the pit. The aluminum alloy matrix under the sand film layer has been corroded, and there is a tendency to gather and peel. As the corrosive solution continues to permeate holes and cracks, the corrosion area expands, resulting in a reduction in the corrosion resistance of the sample. In the enlarged image, the interior of the pitting area is also filled with corrosion products, suggesting that corrosion happens both on the surface and in the holes of the coating. As a result, the shielding effect of the coating is significantly diminished.

Figure 2b shows the microscopic corrosion morphology of a flat powder coating (sample 2#) after 200 h of immersion in the 3.5 wt.% NaCl solution. There are fewer corrosion products, and the majority of the pitting pits are more regularly shaped. A few are irregular holes, and there are fewer corrosion products. There are a small number of cracks and a great number of powder particles on the surface, and the powder particles are not covered by corrosion products, indicating that corrosion is more likely to occur at holes and fractures at the particle interface than on the bulk surface. The 6061 aluminum alloy substrate is not corroded, and the coating provides protection.

Figure 2c shows the microscopic corrosion morphology of a hard anodized aluminum alloy sample (sample 3#) after 200 h of immersion in the 3.5 wt.% NaCl solution. The surface of the coating is covered with many discontinuous particle corrosion products, and the corrosion type is mainly pitting corrosion. The surrounding and internal corrosion pits are covered by corrosion products; some pitting areas have cracks; there is a tendency to peel; the substrate is severely corroded; the degree of pitting corrosion is severe; and the coating has a poor protection effect on the substrate.

Figure 2d shows the microscopic corrosion morphology of ordinary anodized heat-sealed hole aluminum alloy (sample 4#) after 200 h of immersion in the 3.5 wt.% NaCl solution. There are no granular corrosion products on the sample surface, with the enlarged image showing that the sample surface is covered with a dense and continuous corrosion product film.

### 3.2. Electrochemical Impedance Spectroscopy

Figure 3 shows the electrochemical impedance spectrum obtained after the immersion of the sand-grained powder aluminum alloy sample (sample 1#) in a 3.5 wt.% NaCl solution for 2 h and 200 h, respectively. As the immersion time increases to 200 h, the diameter of the capacitor arc decreases significantly, indicating a great decrease in the impedance value. In order to better analyze the corrosion process of the coating, electrochemical data were fitted, and the results are shown in Table 1. Where *R_S_* is the solution resistance, *R*_1_ is the additional resistance of the electrolyte solution in the local corrosion part, and *R_ct_* represents the charge transfer resistance of the electric double layer. The improved corrosion efficiency obtained from EIS measurements was calculated according to the following Equation (1)
(1)$$v\%=\frac{R_{ct}^{0}-R_{ct}}{R_{ct}^{0}}\cdot 100\%$$
where *R_ct_* is the charge-transfer resistance of the sample in the test solution for 200 h and $R_{ct}^{0}$ is the charge-transfer resistance of the sample in the test solution for 2 h. Since the environment of the entire system is not ideal, the capacitor is replaced by a constant phase angle element (*CPE*). The *CPE* has two components, namely *CPE-Y*_0_ and *n*, where *Y*_0_ is the frequency-independent admittance and *n* is the diffusion effect index. When *n* = 1, the *CPE* represents the capacitance; when 0.5 < *n* < 1, it means that the surface of the working electrode has high roughness or a non-uniform current distribution; and when *n* = 0.5, it corresponds to the diffusion process at low frequency or the response of the porous electrode at high frequency [29]. With the extension of immersion time, the value of *CPE-Y*_0_ increases significantly. According to Formula (2) [30]:(2)$$C=\varepsilon \varepsilon_{0}Ad^{-1}$$

The *C* is the capacitance, the *ε* is the permittivity of the oxide layer, the *ε*_0_ is the dielectric constant in the vacuum, the *A* is the surface area of the sample in the experiment, and the *d* is the thickness of the passivation layer. If the surface area *d* of the sample does not change significantly with the increase in immersion time, the increase in *CPE_ox_* can be attributed to the erosion effect of Cl^−^ on the oxide layer of aluminum alloy, which reduces the thickness *d* of the passivation layer. On the other hand, Ferreira et al. [31] found that the oxide layer of aluminum alloy (Al_2_O_3_·xH_2_O) increased due to hydration *ε* after being immersed and corroded. Therefore, the increase in capacitance value may be related to the continuous increase in hydration, which in turn leads to a decrease in the corrosion resistance of the coating on aluminum alloy surfaces. Due to the characteristics of the sand coating’s surface treatment process, its corrosion resistance is drastically reduced by hydration.

Figure 4 shows the EIS results of the flat powder-coated aluminum alloy (sample 2#) immersed in a 3.5 wt.% NaCl solution for 2 h and 200 h, respectively. The fitting data are summarized in Table 2. With the increase in immersion time, the capacitor arc diameter of sample 2# decreased significantly, and the value of *R_ct_* decreased from 451.16 kΩ·cm^2^ to 108.43 kΩ·cm^2^. The value of *CPE_dl_-Y*_0_ increases from 7.976 × 10^−7^ s^n^·Ω^−1^·cm^−1^ to 3.1047 × 10^−5^ s^n^·Ω^−1^·cm^−1^. Improved corrosion efficiency increased by 75.97%. In general, the value of *CPE_dl_* is positively correlated with the quantity and area of pitting. Therefore, with the increase in immersion time, the quantity of pitting corrosion in the coating increases, and the *CPE_dl_* value increases by two orders of magnitude.

Figure 5 shows the EIS results of the hard anodized aluminum alloy (sample 3#) immersed in a 3.5 wt.% NaCl solution for 2 h and 200 h, respectively. The fitting data are summarized in Table 3. It can be seen from Figure 5a that with the extension of the immersion time, the diameter of the capacitor arc of sample 3# decreases significantly, the *R_ct_* value of the electric double layer charge transfer resistance decreases sharply from 635.42 kΩ·cm^2^ to 11.94 kΩ·cm^2^, and the values of *CPE_ox_* and *CPE_dl_* increase by three orders of magnitude, corresponding to the severe pitting corrosion. Improved corrosion efficiency increased by 98.12%. The erosion of chloride ions on the film is the main corrosion behavior, which in turn leads to the formation of local corrosion microbatteries, which is manifested as a decrease in (*R*_1_ + *R_ct_*), and when the corrosion products accumulate near the pit, the erosion of chloride ions will be weakened to a certain extent and *R*_1_ will increase slightly.

In Figure 6b, it can be seen that the Bode polt of sample 4# shows three-time constants after 2 h of corrosion, namely high-frequency, medium-frequency, and low-frequency capacitive arcs as well as low-frequency sensory arcs, indicating that the surface of the sample is still in an activated state at this time, the surface of the specimen is not completely covered by corrosion products, and the aluminum alloy substrate is still actively dissolving. With the increase in immersion time, the high-frequency, medium-frequency, and low-frequency capacitive arcs continue to shrink. And the low-frequency sensory arc resistance disappears, with the sample surface completely covered by the corrosion product film. The fitting data are summarized in Table 4. Improved corrosion efficiency increased by 98.73%.

### 3.3. Polarization Curve

Figure 7 shows the potentiodynamic polarization curves of sample 1# after corrosion in a 3.5 wt.% NaCl solution for 2 h and 200 h. Both the cathode branch and the anode branch show typical Tafel behavior, and the cathode slope (*b_C_*) and anode slope (*b_A_*) and corrosion current density (*I_corr_*) can be calculated by the Tafel extrapolation, as shown in Table 5. The improved corrosion efficiency values obtained from Tafel measurements were calculated according to the following Equation (3):(3)$$\eta \%=\frac{i_{corr}-i_{corr}^{0}}{i_{corr}}\cdot 100\%$$
where $i_{corr}^{0}$ is the corrosion current density of the sample in the test solution for 2 h and *i_corr_* is the corrosion current density of the sample in the test solution for 200 h. After 2 h of immersion, several peeks occur in the current density plot when the applied potential is higher than *E_corr_*, indicating the pitting corrosion begins and the surface of the coating is activated. With the immersion time extending, the corrosion potential of sample 1# decreased from −350.97 mV to −486.49 mV. The corrosion current increased from 0.1709 μA cm^−2^ to 4.0059 μA cm^−2^, and the *b*_A_ value increased. With an increase in immersion time, the anode reaction exhibited the primary control effect, and the sample exhibited obvious passivation. Improved corrosion efficiency increased by 95.73%.

Figure 8 shows the potentiodynamic polarization curves of sample 2# after corrosion in a 3.5 wt.% NaCl solution for 2 h and 200 h. The fitting results are shown in Table 6. The cathode curve and the anode curve exhibit typical Tafel behavior when immersed for 200 h. When immersed for 2 h, the anode curve does not show an obvious Tafel behavior, indicating that the pit potential *E_pit_* is approximately equal to *E_corr_* at this time, and the sample 2# coating corrodes under the self-corrosion potential. The corrosion potential decreased from −401.15 mV to −471.05 mV, and the corrosion current increased from 0.2418 μA cm^−2^ to 5.4439 μA cm^−2^. Both the cathodic and anodic reactions affect the surface electrochemical behavior of the sample, and anodic polarization acts as the main control mechanism. Improved corrosion efficiency increased by 95.56%.

Figure 9 shows the potentiodynamic polarization curves of sample 3# after corrosion in a 3.5 wt.% NaCl solution for 2 h and 200 h. The fitting results are shown in Table 7. With the extension of immersion time, the corrosion potential of the hard anodized coating decreases significantly, from −416.36 mV to −754.58 mV, and the corrosion current increases from 0.1371 μA cm^−2^ to 1.4051 μA cm^−2^. Improved corrosion efficiency increased by 90.24%. After being immersed for 2 h, the cathodic polarization is obvious, while the anodic polarization is slow. The substrate slowly dissolves and begins to pit, followed by passivation. When immersed for 200 h, the cathodic polarization process slows down and the anodic polarization process dominates. The corrosion rate of the sample is accelerated in this case, and the corrosion resistance seriously decreases.

Figure 10 shows the potentiodynamic polarization curves of sample 4# after corrosion in a 3.5 wt.% NaCl solution for 2 h and 200 h. The fitting results are shown in Table 8. The corrosion current first decreases slowly with the increase in potential and reaches a minimum value at *E_corr_*. Then the corrosion current increases rapidly with the small increase in potential. The *E_pit_* and *E_corr_* are closer to the generation of pitting corrosion, and the sample is passivated as the potential continues to rise. When the immersion time reaches 200 h, the corrosion potential decreases from −412.16 mV to −559.81 mV, and the corrosion current increases from 0.3069 μA cm^−2^ to 2.8404 μA cm^−2^, with both cathodic polarization and anodic polarization exhibiting typical Tafel behavior. Improved corrosion efficiency increased by 89.19%.

### 3.4. Summary

The electrochemical impedance spectra of aluminum alloy coatings with different surface treatment processes are summarized in Figure 11 and Table 9. When immersed for 2 h, the corrosion resistance order of the four samples is: 1# > 3# > 2# > 4#. Compared with sample 1#, the improved corrosion efficiency is increased by 58.65% (sample 2#), 41.77% (sample 3#), and 79.33% (sample 4#), respectively. When immersed for 200 h, the corrosion resistance order of the four samples is: 1# > 2# > 3# > 4#. Compared with sample 1#, the improved corrosion efficiency is increased by 39.48% (sample 2#), 93.33% (sample 3#), and 98.39% (sample 4#), respectively.

Under the double-layer protection of large particles and small powder, the coating of sample #1 is relatively dense, with a smaller porosity and obviously better corrosion resistance than the other three tested samples. It can be seen using the scanning electron microscope that the surface holes of sample #4 are not completely sealed, and there are many other surface defects that also offer the possibility of pitting corrosion. Due to the treatment process, the greater the thickness of sample #4’s oxide film, the more defects in appearance, the more cracks, and the poorer the uniformity and continuity of the oxide film’s surface. Therefore, in the early stages of corrosion, hard anodized coatings can also exhibit good corrosion resistance, but as immersion time is extended, the corrosion resistance of the sample decreases dramatically.

The polarization curves of different coated samples are summarized in Figure 12 and Table 10. When immersed for 2 h, the corrosion resistance order is 3# > 1# > 2# > 4#. Compared with sample 1#, the improved corrosion efficiency is increased by 41.49% (sample 2#) and 79.58% (sample 4#), respectively. While the inhibition efficiency is increased by 19.78% (sample 3#). From the corrosion potential observation of four coatings, sample 1# has the highest corrosion potential, indicating the best corrosion resistance. When immersed for 200 h, the corrosion resistance order becomes 3# > 4# > 1# > 2#. Compared with sample 1#, the inhibition efficiency is increased by 64.92% (sample 3#) and 29.09% (sample 4#), respectively. While the improved corrosion efficiency is increased by 35.90% (sample 2#). The corrosion potentials of the four samples move in the negative direction; samples 1# and 2# have the highest corrosion potentials, while sample 3#’s corrosion potential drops the most.

As for the electrochemical protection performance of the coating on aluminum alloy observed from the cathodic polarization, the hard anodized cathodic polarization curve of the sample 3# was the steepest when immersed for 2 h, indicating that it had good polarization performance, and the ordinary heat-sealing anodized sample (sample 4#) had the worst polarization performance and the flattest curve slope. When the immersion time reaches 200 h, the cathodic polarization curve of hard anodized sample 3# has the flattest slope, the worst polarization performance, and the weakest protection effect on the substrate.

## 4. Conclusions

In this work, the influence of four treated coatings on aluminum alloy profile coating surfaces on corrosion resistance was studied by static corrosion immersion tests and electrochemical tests, and the main conclusions are as follows: (1) When corroded in a 3.5 wt.% NaCl solution for 2 h, the corrosion inhibition performances of samples 2# and 4# are poor while the inhibition performances of samples 1# and 3# are good, and the inhibition performance follows the sequence: 1# > 3# > 2# > 4#. (2) When corroded in a 3.5 wt.% NaCl solution for 200 h, the corrosion inhibition performances of samples 1# and 2# are poor while the inhibition performances of samples 3# and 4# are good, and the inhibition performance follows the sequence: 3# > 4# > 1# > 2#. (3) With the increase in corrosion time, the corrosion inhibition performance of the four coated aluminum alloy materials decreased significantly, and the order of decline is 1# > 3# > 2# > 4#.

## Figures and Tables

**Figure 1 materials-16-06027-f001:**
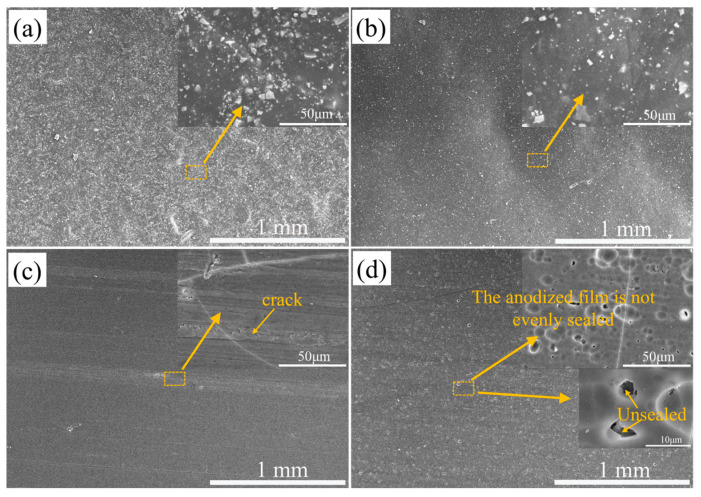
Pre-corrosion micromorphology of aluminum alloy samples with different surface treatments. (**a**) sand powder coating, (**b**) flat powder coating, (**c**) hard anodizing film, and (**d**) ordinary anodized heat seal hole specimen.

**Figure 2 materials-16-06027-f002:**
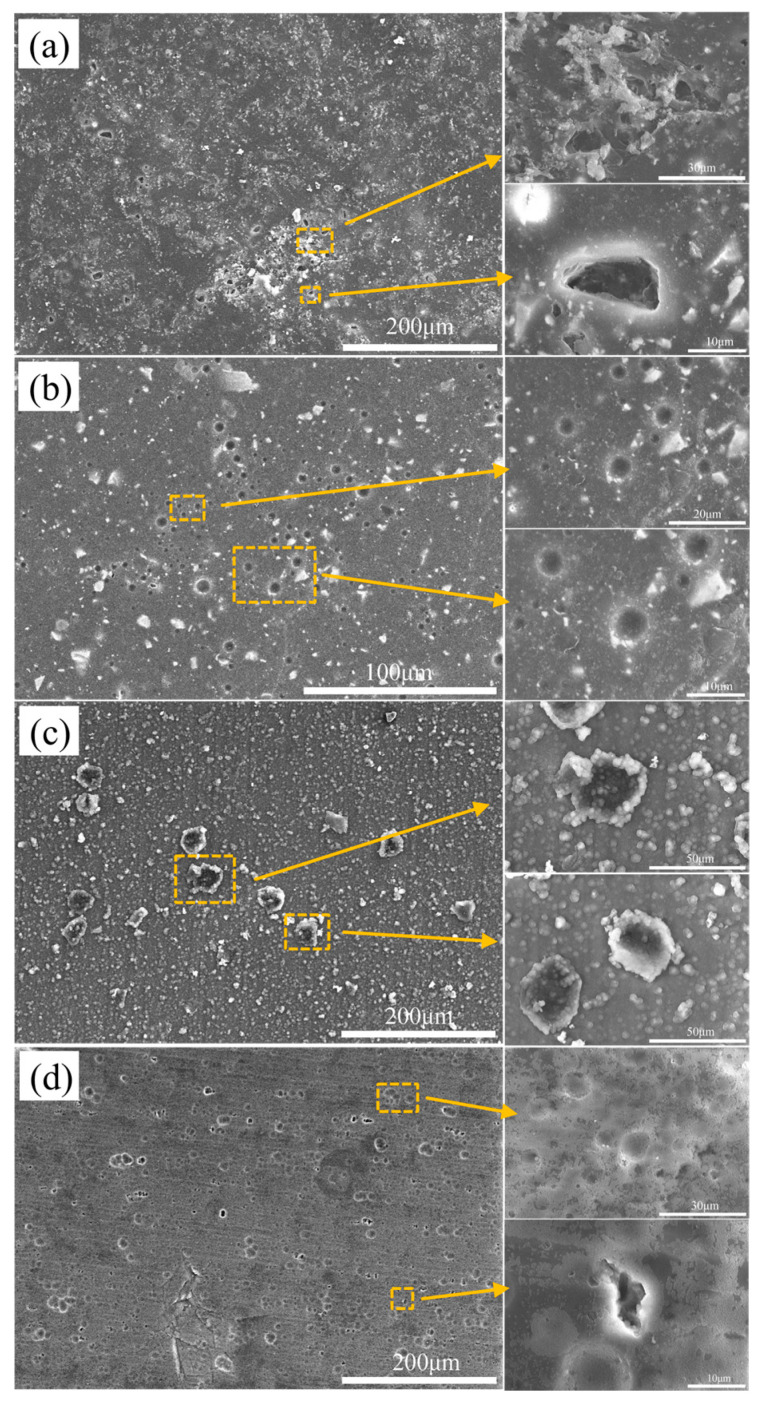
Micromorphology of different treated samples after corrosion in 3.5 wt.% NaCl solution for 200 h. (**a**) sand powder coating, (**b**) flat powder coating, (**c**) hard anodizing film, and (**d**) ordinary anodized heat seal hole specimen.

**Figure 3 materials-16-06027-f003:**
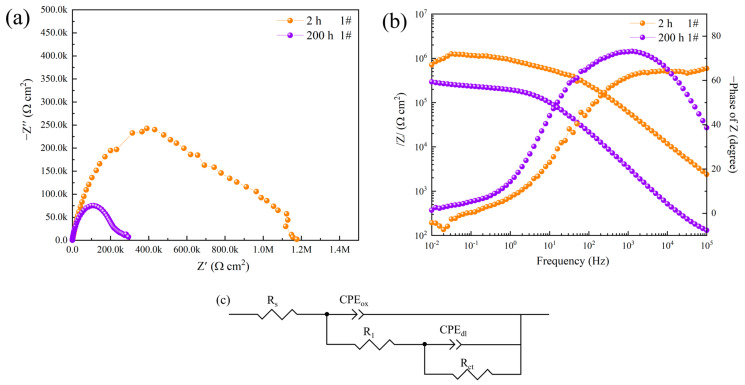
Nyquist plot (**a**), Bode impedance plot (**b**), and equivalent circuit (**c**) of sample 1# after corrosion in 3.5 wt.% NaCl solution for 2 h and 200 h.

**Figure 4 materials-16-06027-f004:**
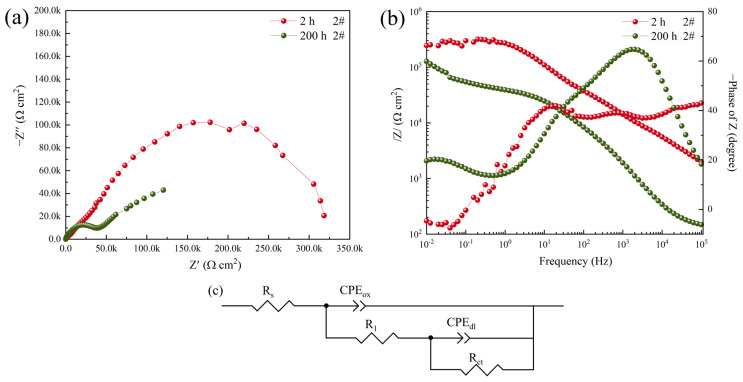
Nyquist plot (**a**), Bode impedance plot (**b**), and equivalent circuit (**c**) of sample 2# after corrosion in 3.5 wt.% NaCl solution for 2 h and 200 h.

**Figure 5 materials-16-06027-f005:**
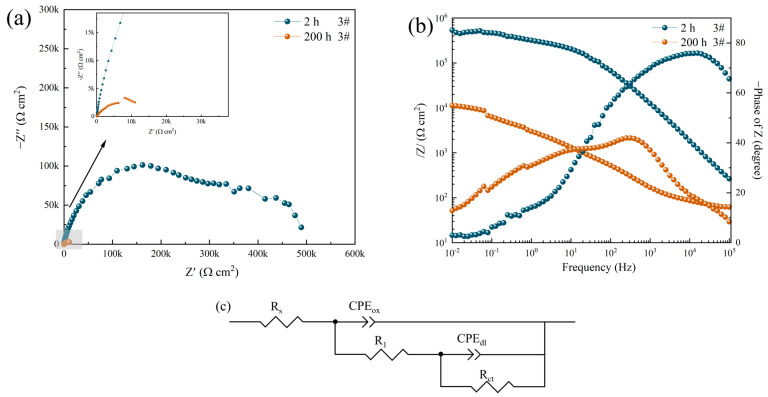
Nyquist plot (**a**), Bode impedance plot (**b**), and equivalent circuit (**c**) of sample 3# after corrosion in 3.5 wt.% NaCl solution for 2 h and 200 h.

**Figure 6 materials-16-06027-f006:**
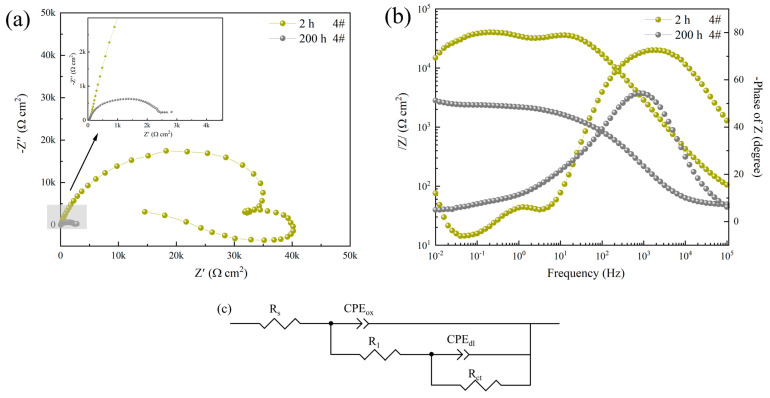
Nyquist plot (**a**), Bode impedance plot (**b**), and equivalent circuit (**c**) of sample 4# after corrosion in 3.5 wt.% NaCl solution for 2 h and 200 h.

**Figure 7 materials-16-06027-f007:**
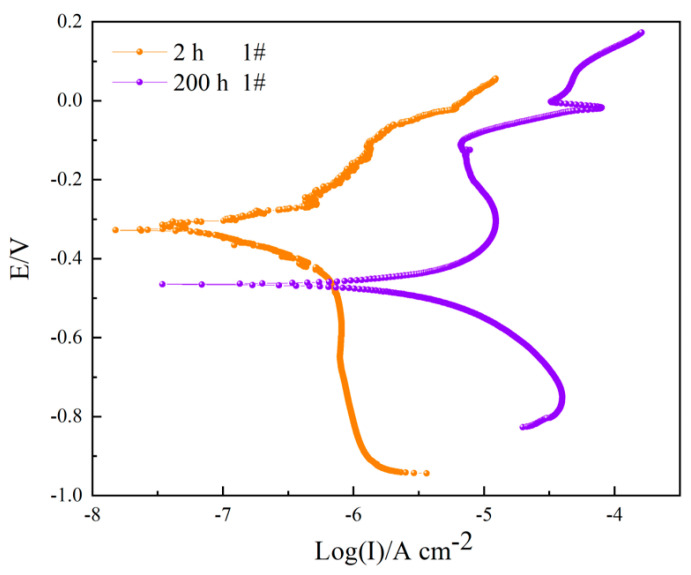
Potentiodynamic polarization curves of sample 1# after corrosion in 3.5 wt.% NaCl solution for 2 h and 200 h.

**Figure 8 materials-16-06027-f008:**
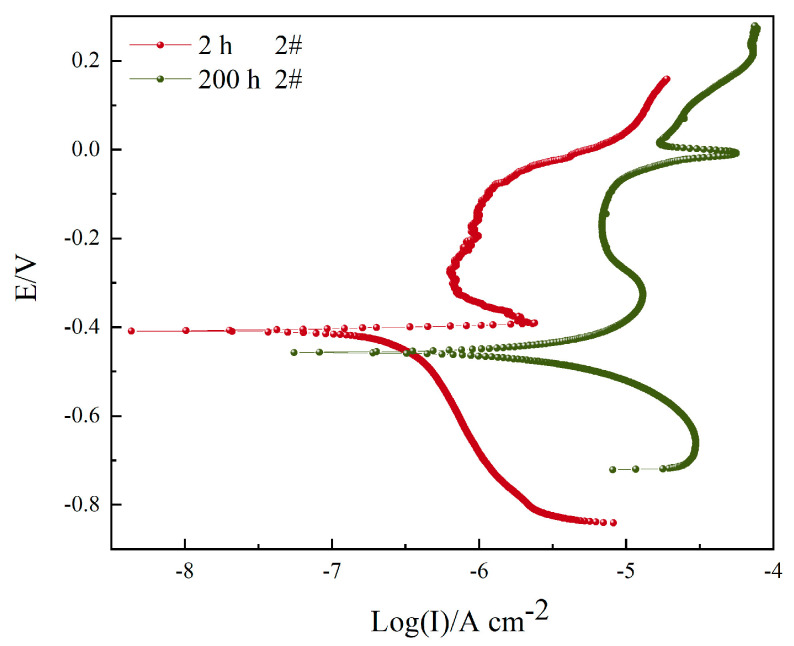
Potentiodynamic polarization curves of sample 2# after corrosion in 3.5 wt.% NaCl solution for 2 h and 200 h.

**Figure 9 materials-16-06027-f009:**
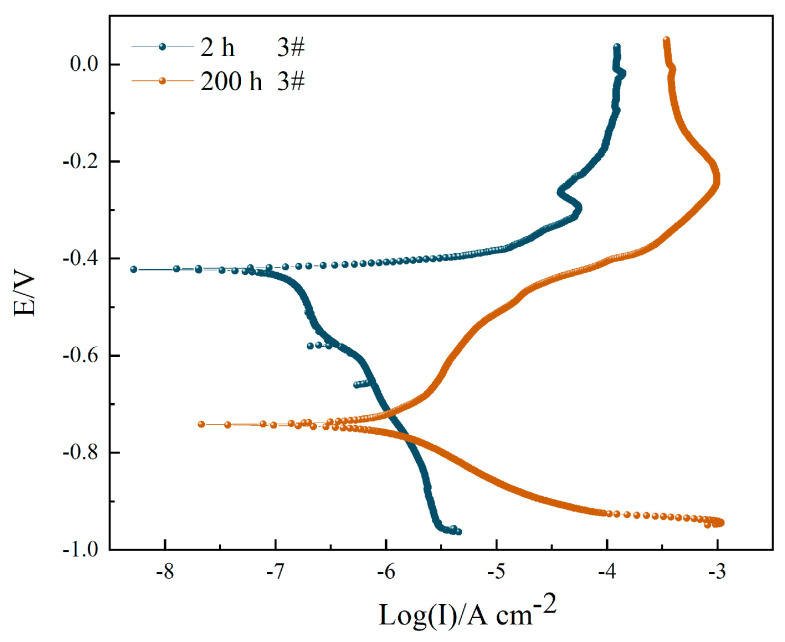
Potentiodynamic polarization curves of sample 3# after corrosion in 3.5 wt.% NaCl solution for 2 h and 200 h.

**Figure 10 materials-16-06027-f010:**
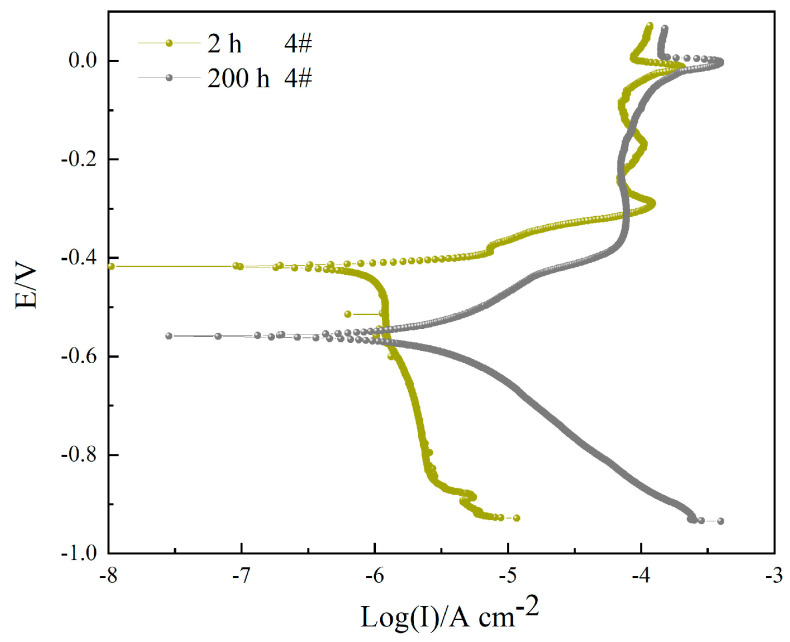
Potentiodynamic polarization curves of sample 4# after corrosion in 3.5 wt.% NaCl solution for 2 h and 200 h.

**Figure 11 materials-16-06027-f011:**
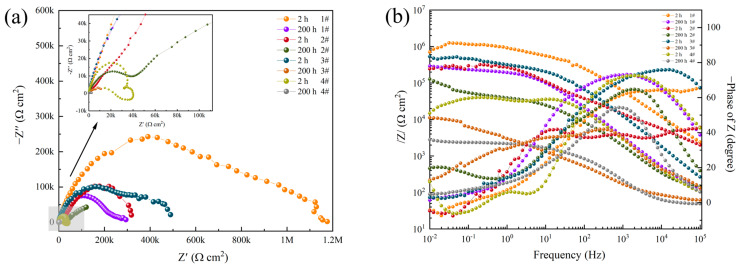
Nyquist plots (**a**) and Bode impedance plots (**b**) of different samples after corrosion in 3.5 wt.% NaCl solution for 2 h and 200 h.

**Figure 12 materials-16-06027-f012:**
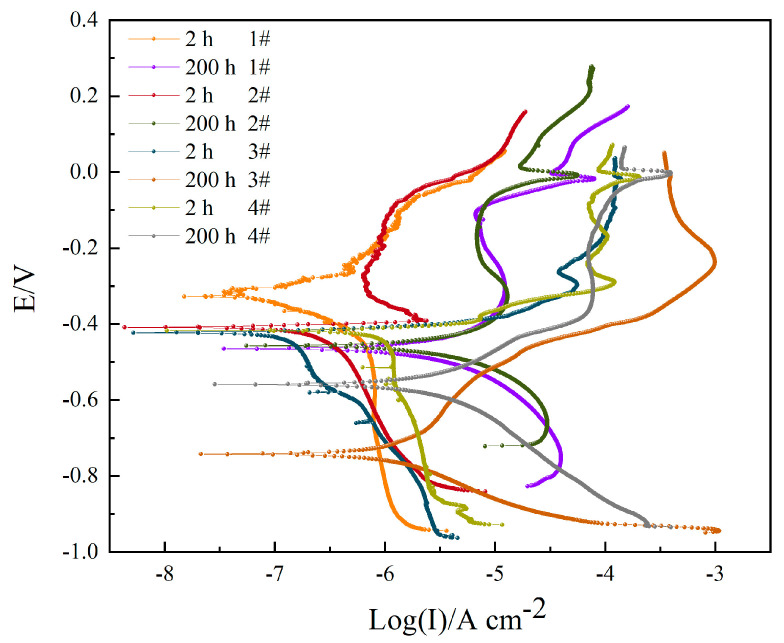
Potentiodynamic polarization curves of different samples after corrosion in 3.5 wt.% NaCl solution for 2 h and 200 h.

**Table 1 materials-16-06027-t001:** The fitting results of sample 1# from the electrochemical tests.

Sample	*R_s_* /(Ω·cm^2^)	*CPE_ox_-Y*_0_ /(s^n^·Ω^−1^·cm^−1^)	*CPE-n_ox_*	*R*_1_ /(Ω·cm^2^)	*CPE_dl_-Y*_0_ /(s^n^·Ω^−1^·cm^−1^)	*CPE-n_dl_*	*R_ct_* /(kΩ·cm^2^)	*v* (%)
1#, 2 h	99.53 ± 6.23	2.1170 × 10^−9^	0.9283	8.0740 × 10^4^	9.0023 × 10^−8^	0.5434	1091.20 ± 50.12	-
1#, 200 h	83.20 ± 4.30	2.1803 × 10^−7^	0.8246	1.4101 × 10^6^	5.0555 × 10^−6^	0.2777	179.15 ± 2.44	83.58

**Table 2 materials-16-06027-t002:** The fitting results of sample 2# from the electrochemical tests.

Sample	*R_s_* /(Ω·cm^2^)	*CPE_ox_-Y*_0_ /(s^n^·Ω^−1^·cm^−1^)	*CPE-n_ox_*	*R*_1_ /(Ω·cm^2^)	*CPE_dl_-Y*_0_ /(s^n^·Ω^−1^·cm^−1^)	*CPE-n_dl_*	*R_ct_* /(kΩ·cm^2^)	*v* (%)
2#, 2 h	133.2 ± 10.2	1.4130 × 10^−9^	0.9543	3.8050 × 10^3^	7.976 × 10^−7^	0.5412	451.16 ± 23.87	-
2#, 200 h	116.1 ± 7.3	6.7045 × 10^−7^	0.7699	2.5064 × 10^5^	3.1047 × 10^−5^	0.3362	108.43 ± 2.17	75.97

**Table 3 materials-16-06027-t003:** The fitting results of sample 3# from the electrochemical tests.

Sample	*R_s_* /(Ω·cm^2^)	*CPE_ox_-Y*_0_ /(s^n^·Ω^−1^·cm^−1^)	*CPE-n_ox_*	*R*_1_ /(Ω·cm^2^)	*CPE_dl_-Y*_0_ /(s^n^·Ω^−1^·cm^−1^)	*CPE-n_dl_*	*R_ct_* /(kΩ·cm^2^)	*v* (%)
3#, 2 h	49.39 ± 3.4	3.6644 × 10^−8^	0.8685	1.410 × 10^3^	8.3306 × 10^−7^	0.2883	635.42 ± 10.15	-
3#, 200 h	55.91 ± 2.6	4.5205 × 10^−5^	0.5788	2.512 × 10^3^	1.2077 × 10^−4^	0.5467	11.94 ± 0.95	98.12

**Table 4 materials-16-06027-t004:** The fitting results of sample 4# from the electrochemical tests.

Sample	*R_s_* /(Ω·cm^2^)	*CPE_ox_-Y*_0_ /(s^n^·Ω^−1^·cm^−1^)	*CPE-n_ox_*	*R*_1_ /(Ω·cm^2^)	*CPE_dl_-Y*_0_ /(s^n^·Ω^−1^·cm^−1^)	*CPE-n_dl_*	*R_ct_* /(kΩ·cm^2^)	*v* (%)
4#, 2 h	62.41 ± 7.48	2.3807 × 10^−7^	0.8342	4.2038 × 10^4^	1.8558 × 10^−5^	1.543	225.5 ± 3.4	-
4#, 200 h	47.17 ± 6.26	2.534 × 10^−6^	0.8523	1.044 × 10^3^	7.6539 × 10^−5^	0.2938	2.87 ± 0.22	98.73

**Table 5 materials-16-06027-t005:** Potentiodynamic polarization parameters of sample 1# after corrosion in 3.5 wt.% NaCl solution for 2 h and 200 h.

Sample	*I_corr_*/(μA cm^−2^)	*E*_0_ (mV)	−*b_C_*/(mV dec^−1^)	*b_A_*/(mV dec^−1^)	*η* (%)
1#, 2 h	0.1709 ± 0.0422	−350.97 ± 7.56	160.20	226.68	-
1#, 200 h	4.0059 ± 0.1130	−486.49 ± 4.50	163.94	298.19	95.73

**Table 6 materials-16-06027-t006:** Potentiodynamic polarization parameters of sample 2# after corrosion in 3.5 wt.% NaCl solution for 2 h and 200 h.

Sample	*I_corr_*/(μA cm^−2^)	*E*_0_ (mV)	−*b_C_*/(mV dec^−1^)	*b_A_*/(mV dec^−1^)	*η* (%)
2#, 2 h	0.2418 ± 0.0330	−401.15 ± 5.24	337.00	9.08	-
2#, 200 h	5.4439 ± 0.2503	−471.05 ± 4.16	176.98	331.61	95.56

**Table 7 materials-16-06027-t007:** Potentiodynamic polarization parameters of sample 3# after corrosion in 3.5 wt.% NaCl solution for 2 h and 200 h.

Sample	*I_corr_*/(μA cm^−2^)	*E*_0_ (mV)	−*b_C_*/(mV dec^−1^)	*b_A_*/(mV dec^−1^)	*η* (%)
3#, 2 h	0.1371 ± 0.0214	−416.36 ± 4.88	532.73	11.25	-
3#, 200 h	1.4051 ± 0.0832	−754.58 ± 9.00	125.20	323.83	90.24

**Table 8 materials-16-06027-t008:** Potentiodynamic polarization parameters of sample 4# after corrosion in 3.5 wt.% NaCl solution for 2 h and 200 h.

Sample	*I_corr_*/(μA cm^−2^)	*E*_0_ (mV)	−*b_C_*/(mV dec^−1^)	*b_A_*/(mV dec^−1^)	*η* (%)
4#, 2 h	0.3069 ± 0.0412	−412.16 ± 3.10	69.22	10.02	-
4#, 200 h	2.8404 ± 0.1004	−559.81 ± 5.24	173.79	165.02	89.19

**Table 9 materials-16-06027-t009:** The fitting results of the different samples from the electrochemical tests.

Sample	*R_s_* /(Ω·cm^2^)	*CPE_ox_-Y*_0_ /(s^n^·Ω^−1^·cm^−1^)	*CPE-n_ox_*	*R*_1_ /(Ω·cm^2^)	*CPE_dl_-Y*_0_ /(s^n^·Ω^−1^·cm^−1^)	*CPE-n_dl_*	*R_ct_* /(kΩ·cm^2^)	*v* (%)
2 h 1#	995.03	2.117 × 10^−9^	0.9283	8074	9.0023 × 10^−8^	0.5434	1091.2	-
2 h 2#	1343.2	1.413 × 10^−9^	0.9543	3805	7.976 × 10^−7^	0.5412	451.16	58.65
2 h 3#	49.39	3.6644 × 10^−8^	0.8685	1410	8.3306 × 10^−7^	0.2883	635.42	41.77
2 h 4#	62.41	2.3807 × 10^−7^	0.8342	4.2038 × 10^4^	1.8558 × 10^−5^	1.543	225.5	79.33
200 h 1#	83.2	2.1803 × 10^−7^	0.8246	141,010	5.0555 × 10^−6^	0.2777	179.15	-
200 h 2#	116.1	6.7045 × 10^−7^	0.7699	25,064	3.1047 × 10^−5^	0.3362	108.43	39.48
200 h 3#	55.91	4.5205 × 10^−5^	0.5788	2512	1.2077 × 10^−4^	0.5467	11.94	93.33
200 h 4#	47.17	2.534 × 10^−6^	0.8523	10.44	7.6539 × 10^−5^	0.2938	2.87	98.39

**Table 10 materials-16-06027-t010:** Potentiodynamic polarization parameters of different samples after corrosion in 3.5 wt.% NaCl solution for 2 h and 200 h.

Sample	*I_corr_/*(μA cm^−2^)	*E_0_* (mV)	−*b_C_*/(mV dec^−1^)	*b_A_*/(mV dec^−1^)	*η* (%)
2 h 1#	0.1709	−350.97	160.20	226.68	-
2 h 2#	0.2418	−401.15	337.00	9.08	41.49
2 h 3#	0.1371	−416.36	532.73	11.25	−19.78
2 h 4#	0.3069	−412.16	69.22	10.02	79.58
200 h 1#	4.0059	−486.49	163.94	298.19	-
200 h 2#	5.4439	−471.05	176.98	331.61	35.90
200 h 3#	1.4051	−754.58	125.20	323.83	−64.92
200 h 4#	2.8404	−559.81	173.79	165.02	−29.09

## Data Availability

Not applicable.

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
