# Peer review of "Study on the Influence of Surface Treatment Process on the Corrosion Resistance of Aluminum Alloy Profile Coating"

_materials, 2023, doi:10.3390/ma16176027_

Round 1

Reviewer 1 Report

The presented manuscript includes a study on the Influence of Surface Treatment Process on the Corrosion Resistance of Aluminum Alloy Profile Coating. 

The results of the work are presented on a good level but some questions and weak points should be mentioned. The paper fits the Materials journal and is of interest.

1. Abstract should include exact results with exact numbers.

2. Introduction. Should be elaborated more in terms of why such treatment is interesting for investigation and what is done and described in the literature. By this, you should show and prove the actuality and novelty of your work. Now this vital part is missed. Why is this research worth investigation taking into account that any commonly used organic coating gives a much bigger Zmod (10^10) compared with shown in this paper (10^6)? 

3. Would be nice to add results with not treated sample (bare AA alloy): SEM, ElChem.

4. Please double-check the whole text for typos, e.g. Fig.10 title, should be sample #3.

5. Please merge tables 1-4 in one table, tables 5-8 in one table, figs 4-7 in one fig, figs 8-11 in one fig, as you nicely did in subsection 3.4.

6. What is the difference between figs 4-11 and fig 13? And tables 1-8 and table 10?. 

7. Please provide a conclusion with exact numbers.

Reviewer 2 Report

Review for materials-2530008

“Study on the Influence of Surface Treatment Process on the 2 Corrosion Resistance of Aluminum Alloy Profile Coating”

.  

1.  Generally, the language and structure of the article needs extensive proofreading.

e.g. This work focus    This work focuses

2. The abstract does not reflect the article's work, and its language is not that good.

3. Figure 1   does not mean anything, there is no morphology as claimed to appear in the images.

4. The captions of all figures are not well written.

5. The summary (3.4) is very long and tedious. This is not a summary.

6. Authors should decide on using either a,b,c,d for the different samples or using the numbers (#).. in the whole manuscript.

7. Inhibition efficiency percentages must be calculated and reported in the text and tables and then discussed to give an idea about the importance of the coating. 

Needs work and improvement

Reviewer 3 Report

1.      Abstract: focus focuses

2.      Abstract: Aluminium alloy should be mentioned

3.      Abstract: electrochemical workstation → you should mention the corrosion methods that you used

4.      Abstract: You are measuring the corrosion of a coated aluminium not only of the coating.

5.      Introduction: Introduction is very limited. There is no discussion on the corrosion performance of coated aluminium and current state-of-art. The added value and novelty of this work should be made clear.

6.      Experimental part: No information on the coatings is given. What was their composition, thickness, morphology, roughness etc.?

7.      Experimental part: Sample preparation prior to corrosion tests is missing.

8.      Experimental part: Repeatability of tests is not defined.

9.      Results and discussion: How can you evaluate thickness from surface images?

10.   Results and discussion: ‘The pitting pit is shallow’. You cannot assess depth based on 2D surface pictures.

11.   Results and discussion: ‘substrate is not severely eroded’ Erosion is a completely different mechanism. Why do you expect to have erosion especially on the substrate? Also ‘pitting pit’ is not correct as one implies the other.

12.   Results and discussion: Corrosion analysis of immersion samples is just a description of SEM images. Discussion on corrosion mechanism, link between corrosion behaviour and microstructure and comparison among the different samples is missing.

13.   Results and discussion: Similarly, there is no critical analysis and discussion of the obtained electrochemical data. Why do these materials behave in a different manner?

14.   Results and discussion: How repeatable are these results? Experimental deviation should be mentioned, unless these are single tests. In the later case, this should be made clear in the text.

15.   Results and discussion: This journal focuses strongly on ‘Materials’, but the material aspect is not well addressed. For example, throughout the text the aluminium alloy type is not even mentioned.  

See the above.

Round 2

Reviewer 1 Report

all comments were addressed 

Reviewer 2 Report

Figure 1: still not clear. Improve.

IE% concept is misunderstood. It is suppose to increase and have a positive value. See text where authors claim in many locations that IE% decreases.

Nombers or letters must be used for different samples. Not both. This is confusing.

English language and flow of ideas are still not to the level of publication in materials.

Needs improving.

Reviewer 3 Report

Dear authors,

After reading the updated version of the article and your point-by-point reply to my comments, I now feel that the manuscript has improved. However, to my point of view, there is one major issue that still remains. This is the fact that you do not provide any information on the materials. By doing so you cannot explain WHY there are differences between these coated materials and thus your manuscript is more of a technical description of the corrosion behaviour of different commercial coatings. As I mentioned before, this journal focuses on ‘Materials’, but the material aspect is not sufficiently discussed. In addition, since you performed triplicate tests, you should add the experimental deviation in results.

Dear Editor,

My main concern with this work is that no information about these coatings are given. Thus, no concrete conclusions on why they behave in a different way can be made. The authors mention that these are commercial coatings, and that the manufacturer cannot provide any details due to confidentiality. However, this journal focuses on Materials.

If you believe that there is sufficient material-based analysis to justify this publication to Materials, please proceed.

Kind regards            
